# APCI Evaluation Method for Cement Concrete Airport Pavements in the Scope of Air Operation Safety and Air Transport Participants Life

**DOI:** 10.3390/ijerph17051663

**Published:** 2020-03-04

**Authors:** Mariusz Wesołowski, Paweł Iwanowski

**Affiliations:** Air Force Institute of Technology, 01-494 Warsaw, Poland; pawel.iwanowski@itwl.pl

**Keywords:** transport, aircraft operations, safety, human life, airfield pavements, technical condition, deterioration

## Abstract

Many factors have an impact on flight operation safety and air transport participants life. This article presents one of them, which is maintenance of the airport infrastructure in a good condition, with proper infrastructure management, in particular of cash and human resources. At the beginning of the article, attention is paid to the aspect of safety and human life in air transport. Also, an overview of world experience in the field of assessment of the technical condition of airport pavements was presented, including the standard method of the Pavement Condition Index (PCI) estimation. Then, the authors propose an innovative method of assessing the condition of airport pavements based on the Airfield Pavement Condition Index (APCI), taking into account, apart from the extent of surface damage, such parameters as load capacity, evenness, roughness, and bond strength. This approach gives a broader picture of the actual condition of the airport pavement, which has a great importance on flight operation safety, including passengers and cruel life. Next the described research method is experimentally verified in real conditions at Polish airports. Finally, an example of using the APCI method in the assessment of selected airport pavements from Polish airports is presented. The results of tests performed on five functional elements of a military airfield are presented. A satisfactory result is obtained for three elements, an adequate—for two.

## 1. Introduction

Air transport is the most modern and dynamically developing branch of transport, which has recorded a several fold increase in the number of operations worldwide over the last decade. The number of travelers is increasing from year to year. Despite the growing attention to the safety of air operations, accidents still happen. Unfortunately, in air transport, an accident usually means fatalities. In the past, aviation events have occurred due to the intake of foreign bodies originating, among others, from airport pavements. That is why it is so important to maintain airport surfaces in the best technical condition.

Proper airport management is a key factor that has a direct impact on the safety of flight operations and air transport participants life. The management of airport functional elements (AFE) shall be based on reliable information on the pavement’s surface condition obtained systemically. This approach enables rational planning of airport pavement repairs and renovations. Many countries’ experiences confirm that proper management of both airport [1] and road [2,3,4,5] infrastructure must rely on detailed and up-to-date information on the condition of a pavement’s surface.

Information about the current condition of the pavement, as well as the ability to predict and forecast the technical condition of the pavement in the future, plays an important role in airport infrastructure management. Iranian scientists in the paper [3] presented attempts to develop an alternative method of determining the PCI (Pavement Condition Index) with the use of optimization techniques based on artificial neural networks and genetic programming. The suggested approach may help reliably extrapolate the PCI indicator in the future. The use of the PCI indicator in the estimation of Remaining Service Life (RSL), as a parameter determining current and future condition of the airport pavement [6] may be sn alternative to the approach above. A similar approach was used to assess the condition of road surfaces in Indonesia. Authors in the article [7] applied the ANOVA method to predict RSL based on the PCI indicator. The PCI indicator is used as a tool to support airport infrastructure management in the state of Indiana. A minimum service level (MSL) has been defined for particular categories of airport functional elements that determines the time to take corrective actions within the element in question [8]. This has a great impact not only on safety, but also on the costs incurred. Sharaf et al. [9] already showed that a properly selected range of PCI indicators assessed determines the proper maintenance of the pavement and the costs.

Various methods of determining the pavement condition index (PCI) can be found in literature and operational practice. They are based mainly on the inventory of pavement surface deteriorations. The standard methods used worldwide are the ASTM D5340 [10] and ASTM D6433 [11] published in the American standards, described by Shahin [1] and cited in this article. The others include the PAVER procedure used worldwide. In Yemen [12], the PAVER procedure was used for the process of evaluation. PCI was the main indicator on the basis of which the maintenance and rehabilitation procedure was proposed. PAVER procedure was also the subject of Kirbas and Karasahin’s research [13]. They used it to evaluate the condition of twenty road sections with different volumes of traffic. PAVER procedure is also used by the Virginia Department of Transportation [14,15]. Some methods use parameters other than visual inspection to assess the PCI index. In [16,17], researchers also consider pavement evenness in the form of the International Roughness Index (IRI) parameter, whereas in Kazakhstan [18], the values of pavement deflections as a result of Falling Weight Deflectometer (FWD) measurement, as well as IRI parameter are taken into account. Researchers from India [19] additionally included the pavement roughness parameter, and proposed the Overall Pavement Condition Index (OPCI) as the pavement condition parameter determined according to the author’s procedure. Chen et al. [20] developed a method for calculating the PCI index based on mathematical models including weighting factors for specific types of pavement deterioration. This allows to some extent to automate the process of analyzing results. A similar approach is presented in articles [21,22], where the authors consider the harmfulness of specific deteriorations in the model of the airport pavement deterioration index.

The collection of reliable and current pavement data is the basis for the PCI indicator’s correct assessment. Visual inspection is made by experts, and thus there is a human factor that can affect the final result. In order to ensure quality at the appropriate level, guides have been created on how to conduct inspection and deal with the collected data. Such a document was developed for the purposes of the Federal Highway Administration [2] and the Indiana Department of Transportation [8]. Italians [23] additionally proposed their own extension of the deteriorations catalogue presented in ASTM D6433 [11], thus adapting it to the needs and morphology of road surfaces in Italian cities.

To eliminate the human factor from the data collection process, it strived for complete automation. In the article [5], the authors, as the example, cite the ARAN system enabling automatic road analysis.

The method of determining the pavement condition indicator proposed by the authors is based not only on the airport pavement’s surface deteriorations, but also takes into account measurement of repairs and technical parameters such as load capacity, evenness, roughness, and surface tensile strength. This approach gives a broader picture of the actual condition of the airport pavement.

The main goal of the work is to develop a method for assessing the condition of airport pavements that gives airport services a practical and reliable tool supporting the management of airport functional elements in a sustainable and safe manner.

At the beginning, the standard PCI method is described. Then, an innovative procedure of APCI determination is presented. Next, the example of using APCI method is shown including results of pavement condition evaluation on a real airfield. Finally, a discussion and conclusions are presented.

## 2. Materials and Methods

Currently known methods of assessing the condition of airport pavements are mainly based on indicators describing degree of pavement degradation. In the book intended for engineers, government institutions, and universities [1], the author describes in detail the method for determining PCI developed and used by the U.S. Army Corps of Engineers. It is a standard method used to assess airport and road surfaces, as well as parking lots, by many institutions around the world. These institutions include the Federal Aviation Administration, The U.S. Department of Defence, the American Public Works Aviation, and many more. The PCI determination method for airport and road surfaces has also been standardized and published in the American standards ASTM D5340 and ASTM D6433.

The PCI indicator is a dimensionless number from 0 to 100, where 0 means the surface is completely degraded, while 100 means the surface is in perfect condition. The Figure 1 shows the standard scale of the PCI indicator and the simplified scale. The PCI indicator is determined based on the result of the pavement’s visual inspection in relation to distress type, quantity, and its severity.

In order to inspect airport pavements, each AFE is divided into smaller elements constituting the test sample according to standard instructions. In the case of concrete pavements, a unit sample of the pavement is 20 ± 8 full-size concrete slabs. The asphalt pavement is virtually divided into areas of approximately 460 ± 180 m^2^ (5000 ± 1000 sq ft.). Inventory of deteriorations on each sample can be time consuming, which is associated with the costs incurred for this purpose. The method allows to limit the number of samples to be tested, and thus to reduce costs and inspection time. The disadvantage of this solution is reduction in the quality of the results. Formula (1) specifying the minimum number of samples was developed in order to obtain results at the assumed confidence level.
(1)$$n\;=\;\frac{N\;\times \;s^{2}}{\left(e^{2}/4\right)\left(N-1\right)\;+s{\;}^{2}},$$
where:

*N*—Total number of samples obtained after dividing the element,

*e*—Allowable estimation error PCI,

*s*—Standard deviation of PCI results obtained on single samples.

The next step is to determine the interval at which samples will be selected for evaluation. Thanks to this, the evaluated samples will be evenly distributed in relation to the element. The interval is the ratio of the total number of samples obtained after dividing the element to the minimum number of samples to be inspected. For example, when the total number of samples is 47 and the minimum number of samples is 13, the interval will be 47/13, i.e., 3.6 (rounded to integer-3). Therefore, for deterioration inspection, samples should be selected in order of numbers 3, 6, 9, 12, (...), 45. Despite the possibility of reducing the number of test samples, it is suggested to check at least 50% of the samples at critical points of the airport. However, in terms of safety, it is very reasonable to evaluate each sample. Inspectors write each noticed damage down in accordance with the legend, specifying its type, severity, quantity, and approximate place of occurrence. Data tables containing the type of damage, its harmfulness, its total quantity, and density basing on the collected results are created for each sample. Shahin [1] gives the next steps in the process of determining the PCI indicator for a single sample:1.Determination of deduct values from the deduct value curves for each distress type and severity. Figure 2 shows a typical deduct value calculation curve.


2.Determination of maximum allowable number of deducts (m), using the following formula:*m_i_* = 1 + (9/95)(100 − *HDV_i_*)(2)
where:

*HDV_i_*—Highest individual deduct value for sample unit i.

3.Determination of maximum corrected deduct value (*CDV*):
a.Determination of *q*, the number of deducts with value greater than 5.0b.Determination of total deduct value (*TVD*) (sum of all individual deduct values)c.Determination *CDV* using *q* and *TVD* using correction curves for Portland cement concrete (PCC) surfaced airfield pavementsd.Reduction of the smallest individual deduct value greater than 5.0 to exactly 5.0,e.Repetition of steps a through c until *q* is equal to 1.0.


The largest of the determined *CDV* is the maximum *CDV*.

4.Calculation PCI from the following formula:

*PCI* = 100 − *CDV_max_*,(3)

In the case where the assessed functional element of the airport was divided into samples of the same area, the PCI for the whole element is calculated as the arithmetic mean of the PCI indicators estimated for a single sample. When the sample surfaces are not the same size, the PCI value is calculated as the weighted average of the PCI indicators estimated for individual samples, in which the size of the individual sample surfaces is taken as the weight.

Authors developed a procedure for assessing the technical condition of the AFE surfaces based on the results of pavement’s parameters. The novelty of the proposed method is of consideration for both deterioration and repair inventory. Load capacity, roughness, evenness, and bond strength are also included in the APCI model. Moreover, the main idea of the APCI model assumes that pavement evaluation can be done only if each input parameter is greater that its minimum requirement; otherwise the AFE should not be operated.

## 3. Results

The assumptions for the practices and proposed level of the APCI (Airfield Pavement Condition Index), adapted to airports in Poland are presented below. In addition, estimated APCI values for several selected AFEs are provided, based on real data obtained during surface visual inspections.

### 3.1. Procedure for APCI Evaluation

The procedure was created in order to standardize the procedure of the technical condition of the AFE pavements assessment on the basis of data obtained from various sources.

The technical condition of the pavement is assessed on the basis of field tests and laboratory tests. Field tests include:deteriorations and repair inventory,pavement load capacity assessment based on the results of elastic deflections obtained in the HWD (Heavy Weight Deflectometer) test,pavement roughness assessment,pavement evenness assessment based on the results obtained in the planograph test,surface bond strength by pull-off.

Laboratory tests include:structural tests of concrete,strength tests, including concrete compressive strength and concrete tensile strength,climate tests, including freeze-thaw resistance and resistance to de-icing agents.

The general scheme of procedure is shown in Figure 3.

The main methodology idea is the assessment of individual parameters based on the results obtained. If the parameter’s limit value is not obtained, corrective work should be undertaken and parameters reassessed. When all the tests are completed, the condition of the AFE’s pavement is determined taking into account the results of the field tests. The technical condition of the pavement is assessed taking into account the condition of the pavement and the laboratory tests results. The process of assessing the condition of the AFE’s pavement itself is illustrated in Figure 4. Example of output data using is presented in Figure 5. The plot presents changes of technical condition for five aprons from one of Polish airports based on the APCI index.

#### 3.1.1. Input Data

Model input data are the results of the assessment of load capacity, degradation, roughness, evenness, and concrete tensile strength. Tests and measurements are carried out in accordance with applicable standard methods.

The load-bearing capacity of airport pavement is assessed in accordance with NO-17-A500: 2016 *Airfield and road pavements-Load capacity testing* [24] is the standard method for deflections measurement with a Heavy Weight Deflectometer (HWD) used for airport testing, and is the basis for the analysis. The thickness and stiffness of structural layers, concrete tensile strength, and soil base parameters directly under the AFE structure are taken into account. For this purpose, a full research of the assessed AFE structure is made. The result is the Pavement Classification Number (PCN) indicator and/or the number of aircraft flight operations.

The level of surface deterioration is determined on the basis of a visual inspection of the surface. The deteriorations and repairs inventory is made taking into account type and measurement of each damage. The basic assessed element is a single 5 × 5 m slab. Its location is determined by numbers, which are adopted in accordance with a strictly defined order. Based on the basic element’s (concrete slab) inspection results, the inspected area can be analyzed as a hectometer, which corresponds to an area of 100 m in length and 5 m in width. For easier slab position identification, successive numbers are assigned to bands and rows. The slabs may have various forms of deteriorations, both linear, point and area, which is why each of them is subject to a separate measurement. The measure of area deteriorations in the form of shallow flaking (Ap), deep flaking (Bg), capillaries cracks (Pw), frost cracks (Pm), deep cavities (Ug), and slabs to replacement (Wp) is the degraded pavement area [m^2^]. The measure of linear damages in the form of a slotted crack (Ps) and a wide slotted crack (Psszer) is its length expressed in [m]. The number and length of cracks longer than 25 cm are written down. The number of pieces [pcs] is a measure of point deteriorations and their repairs, e.g., marl spalls (Op) and fractures of corners (Now). Loss of joint selant mass (M) is given in [m], and thresholds (Pr) caused by displacement of slab edge in height [cm]. Slab to replacement is expressed in units of area [m^2^].

Pavement friction is tested in accordance with the defense standard of Poland NO-17-A501: 2015. *Airfield pavements-Friction testing* [25] and requirements described in Annex 14 of the International Civil Aviation Organization ICAO [26], Standard 9137-AN/898 Part 2 Airport Service Manual [27] and in the Advisory Circular FAA 150/5320-12c [28]. The measurement is made with a device for friction coefficient continuous measurement in accordance with the above documents, ensuring the thickness of the water film under the measuring wheel is at least 1 mm. The test can be carried out at a speed of 65 km/h or 95 km/h and the results are compared with the values from appropriate tables in the documents above. Example of such a device is shown in Figure 4 (PG-3.1.3).

Measurements of unevenness of the assessed surfaces are made based on the NO-17-A502: 2015 *Airfield pavements-Evenness testing* [29]. The measuring device measures and records the height of the clearance between the theoretical line connecting the bottom of the device’s wheels and the pavement. Unevenness amplitudes are measured as a function of path increment, every 10 cm of the tested route, thus creating a set of numbers that are sent to the computer. Unevenness is measured with an accuracy of 0.3 mm. The measuring route is divided into 5 m sections (the most common panel dimensions). The pavement’s evenness assessment is reduced to 5 m long road sections assessment. The standard requirements allow surface deviations by the values specified in the standard [29].

The assessment of the strength of the surface layer of concrete pavements is carried out in accordance with PN-EN 1542: 2000 products and systems for the protection and repair of concrete structures. Test methods consist of measurements of bond strength by pull-off [30]. The test consists in sticking a metal disc with a diameter of 50 mm, a previously appropriately drilled, to the test surface. Then, using a specialized device, the disc is pulled off with a constant strength increase. The sought parameter is determined by dividing the maximum force causing the disc to detach from the structure by its surface area.

#### 3.1.2. Process Analysis

An analysis based on the developed model indicator of the cement concrete (*pl.* BC) airport pavement’s condition with the use of input data is made. The indicator value is calculated using the following formula:(4)APCIBC¯=100−(wUBCUBC+wDBCDBC+wSBCSBC+wRBCRBC+wWodBCWodBC)w
where:

*w*_i_—Characteristic weights for the type of parameter,

*w*—Weight sum

*U*—Load bearing capacity

*D*—Deterioration

*S*—Friction

*R*—Evenness

*Wod*—Bond strength by pull-off.

The above-mentioned weights were selected using the expert method and taking into account many years of experience of experts related to airfield pavement constructions. The results of previous studies carried out at airports throughout Poland were also taken into account. The functional elements of both military and civil airports were examined.

#### 3.1.3. Output Data

Obtained APCI values are evaluated according to the criteria of technical condition assessment. The criteria are presented on the following detailed scale:Good (APCI = 100 ÷ 86)—The surface has little or no damages and only requires routine maintenance.Satisfactory (APCI = 85 ÷ 71)—The surface has minor damages that only requires routine maintenance.Adequate (APCI = 70 ÷ 56)—The surface has damages of low and medium harmfulness. Routine and major repairs should be carried out within a short period of time.Poor (APCI = 55 ÷ 41)—The surface has damages of low, medium, and high harmfulness, which probably cause operational problems. Maintenance work should include routine repairs and reconstructions in the near future.Very poor (APCI = 40 ÷ 27)—The surface has mostly medium and high damages, which causes significant maintenance and operational problems. Immediate intensive maintenance and repairs are needed.Serious (APCI = 26 ÷ 12)—The surface usually has high damages, which cause restrictions in its use. Immediate repair is needed.Unfit (APCI = 11 ÷ 0)—Deterioration of pavement has reached a level where safe air operations are no longer possible. Complete reconstruction is necessary.

The above APCI scale can be presented in a simplified way as:Appropriate  APCI = 100 ÷ 71Degraded   APCI = 70 ÷ 56Unsatisfactory APCI = 55 ÷ 0

The limit values of the APCI indicator individual levels were determined on the basis of many years of experience based on the results of research work obtained over the years.

### 3.2. The Results of the Cement Concrete Airport Pavements Evaluation

Assessment of the surface condition based on the method described in Section 3.1 was made on the basis of tests and measurements carried out at one of the Polish military airports. All functional elements of the airport were examined. The article presents several of them, including the runway and four aprons. The measurements were made in a short period of time, thus ensuring satisfactory repeatability conditions.

An inventory of deteriorations and surface repairs was made based on instructions created at the Air Force Institute of Technology. The inventory was made by experts with many years of experience in this field. Measurements of deflections were carried out using an HWD (Heavy Weight Deflectometer) airport deflectometer and then PCN indices were calculated. The thicknesses of the structural layers and the material characteristics of the pavement were identified on the basis of cylindrical samples taken from the pavement. The evenness of the pavement was assessed based on measurements with the modernized P-3z planograph, while the roughness measurements were made with the ASFT T-10 friction tester. The test of the surface layer’s bond strength was carried out with pull-off apparatus.

The presented assessment results relate to five cement concrete functional elements. Results received in tests are shown in Table 1. The charts (Figure 6, Figure 7, Figure 8, Figure 9 and Figure 10) present deterioration, load bearing capacity, evenness, roughness, and bond strength of the surface layer results for the runway (RWY) and four aprons (APRON1-APRON4). However, Figure 11 shows the estimated pavement condition indicator.

According to the analysis above, APRON 1, APRON 3, and RWY surfaces can be classified as having a satisfactory condition. In contrary, APRON 2 and APRON 4 surfaces qualify as adequate condition and will require routine repairs in a short time. The presented results apply to the entire AFE. In order to obtain greater accuracy of the results, the AFE should be divided into smaller elements and the APCI analysis performed again.

## 4. Discussion

Many factors have an impact on operation safety and air transport participants life. One of them is the maintenance of airport infrastructure to be in an appropriate condition, where proper infrastructure management is important, in terms of financial and human resources administration. As global experience shows, airport services are supported in this field by scientists, who provide them with specialized tools, including pavement assessment systems with PCI data sharing. The method designated by PCI is standardized, the procedure is presented in the article. The method was developed for the needs of the US Army by the Corps of Engineers.

Currently, researchers are competing calculation methods, and more importantly, predicting the value of this parameter. New systems are created, not only for surface deteriorations inspection. The new approaches include the IRI index or measurement of elastic deflections with the FWD device. Others suggest adding a friction parameter to the model. Attempts are being made to automate the process of determining and forecasting PCI using artificial neural networks or statistical methods (ANOVA).

In this article, the authors proposed a new method for the airport pavement condition index (APCI) evaluation. The innovative approach takes into account several factors at the same time, including the degree of pavement deteriorations determined on the basis of an inventory of both pavement deteriorations and repairs. In addition, the model includes the pavement’s load bearing capacity, evenness, and roughness. The load bearing capacity is determined with the deflections measurement, in the model as the PCN indicator. Evenness in the model is considered as a defectiveness parameter, while roughness as a friction coefficient. Due to the jet aircrafts maneuvers on airport pavements, it is important to ensure cleanliness on the surface. There must be no loose elements in the Foreign Object Damage (FOD) zones, so any fractures of the surface is a potential threat to aircraft safety, and to people’s lives. In order to prevent and react in advance to excessive surface fracturing, the pavement is controlled with the bond strength by the pull-off test. The proposed model also takes into account the above parameter. Each type of parameter fills in the model with a characteristic weight. The weights were determined by the expert method.

In order to adapt the scale to the conditions at Polish airports, the authors developed a detailed scale of APCI values, according to which the technical condition of the pavement is assessed. The number of levels remained the same as in the standard PCI method, while the limit values changed slightly. In addition, a simplified scale containing three states—adequate, degraded, and unsatisfactory—was developed. The limit values were determined based on many years of experience based on the results of the research work obtained over the years.

The article presents an example of using the APCI method for an airport pavement evaluation. Five AFEs of an active military airport in Poland were assessed, taking into account the degree of degradation, load-bearing capacity, roughness, evenness, and bond strength of concrete. Undoubtedly, pavement degradation had the greatest impact on the final value of the APCI index. APRON 2 and APRON 4 surfaces, which were 25% and 19% degraded, obtained the lowest APCI rates. Despite the fact that APRON 4 had the most degraded pavement, APRON 2 obtained the lowest APCI index of 68. The largest impact on this situation was the result of testing the concrete bond strength, which significantly reduced the final value of APCI. This behavior of the model shows that the condition of the surface, which is visually in good condition, may pose a threat to the aircraft, and thus to the safety of passengers and crew members. Only taking into account the wide spectrum of airport pavement parameters gives a real picture of its condition.

For operational purposes, one APCI indicator for the entire AFE is sufficient, which shows whether it is possible to perform flight operations on a given element or not. For maintenance purposes, attention should be paid to APCI indicators for individual AFE sections. Due to the large size of the elements, they should be divided into smaller parts with dimensions adapted to the needs of maintenance services. For example, the runway can be divided into 100 m sections. This narrowing of the area under consideration allows for a more accurate assessment of the condition of the entire AFE, enabling the planning of repairs from the worst-case areas with an APCI ratio lower than the average APCI for the whole AFE.

## 5. Conclusions

Maintenance of the airport infrastructure in a good condition is a key factor to increase the level of flight safety and protect human lives and health. The tool created by the authors is aimed at supporting airport services in managing the condition of technical airport pavements, enabling rational and effective disposal of public funds intended for the airport infrastructure maintenance.

The article presents sample results of the evaluation of airport pavements’ condition at a Polish facility. Five airport functional elements were analyzed, including the runway and four aprons. Values of each of the parameters assessed and the final APCI for the evaluated airport functional elements are presented. The method proposed by the authors for assessing the condition of airport pavements based on the APCI takes into account not only distresses but also other pavement parameters. In contrast to methods used so far, the APCI method includes both deterioration and repair inventory, load capacity, roughness, evenness, and bond strength. A broad approach to pavement parameters and application of weighting factors are the main advantages of this method, especially with flight operation safety.

At the preceding stage of this work, a methodology for airport pavement degradation assessment was developed, including repairs carried out and deterioration harmfulness. Work is currently underway in order to determine the impact of specific parameters on the APCI model. In the future, there are plana to use artificial neural networks to optimize the model and predict the surface condition in subsequent years of operation. Work is ongoing within a system supporting Polish airport services in pavement condition management.

## Figures and Tables

**Figure 1 ijerph-17-01663-f001:**
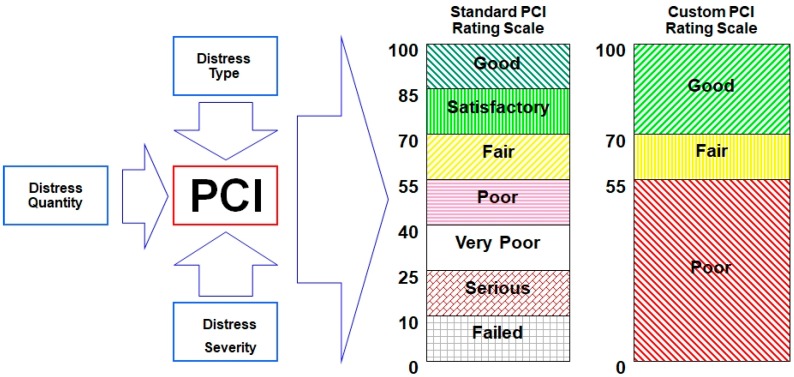
Pavement Condition Index (PCI) rating scale [13].

**Figure 2 ijerph-17-01663-f002:**
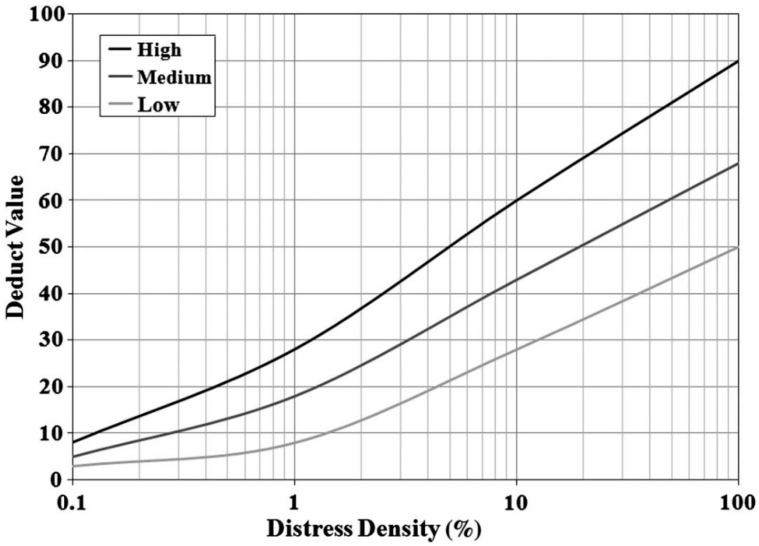
Typical deduct value calculation curve [3].

**Figure 3 ijerph-17-01663-f003:**
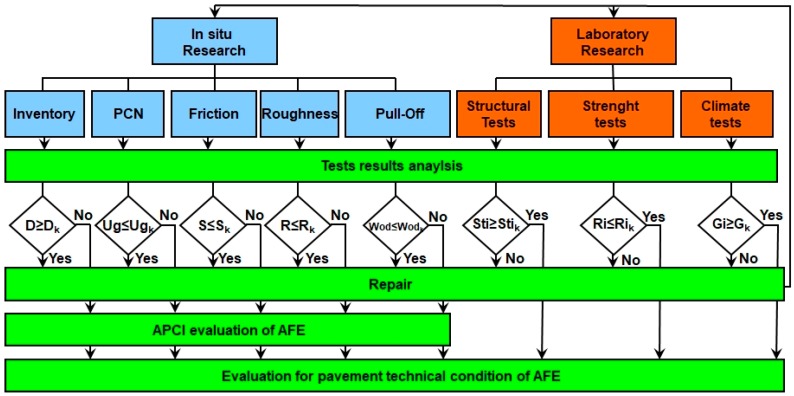
Evaluation of the pavement technical condition of the airport functional elements (AFE). Abbreviations: APCI, Airport Pavement Condition Index; PCN, Pavement Classification Number.

**Figure 4 ijerph-17-01663-f004:**
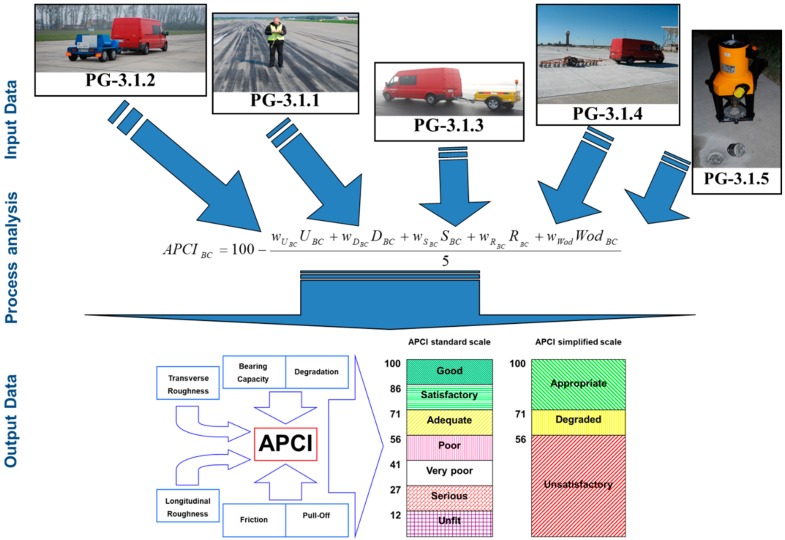
Evaluation of the AFE’s pavement condition.

**Figure 5 ijerph-17-01663-f005:**
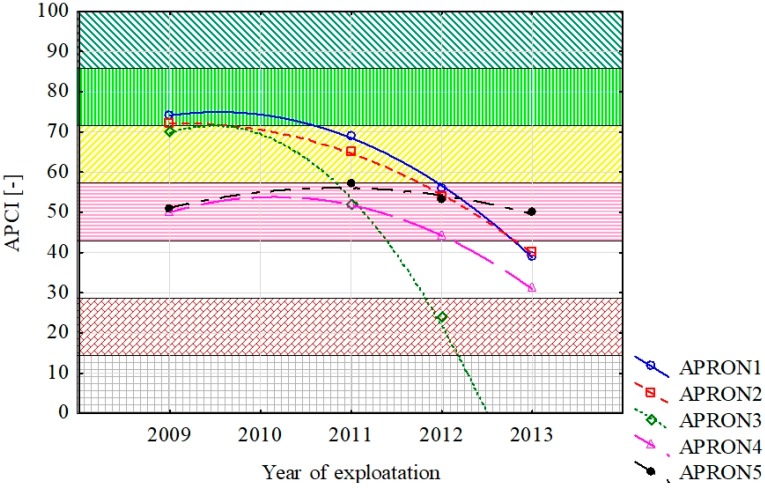
Example of Output Data using.

**Figure 6 ijerph-17-01663-f006:**
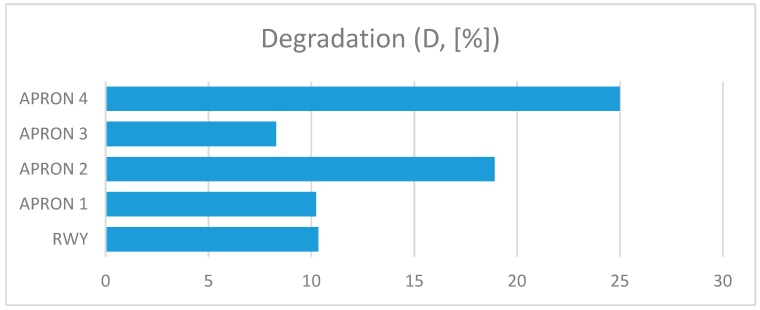
Degradation of AFE’s.

**Figure 7 ijerph-17-01663-f007:**
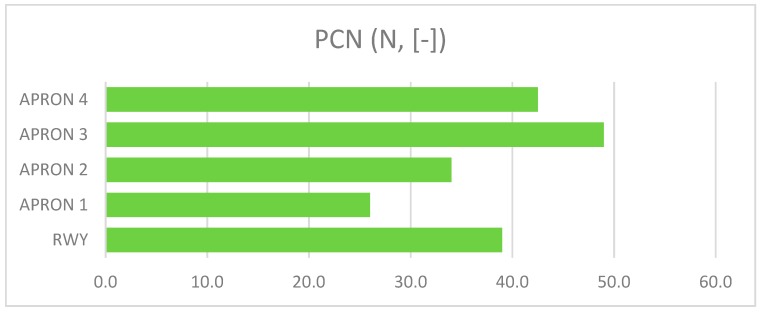
Pavement Classification Number of AFE’s.

**Figure 8 ijerph-17-01663-f008:**
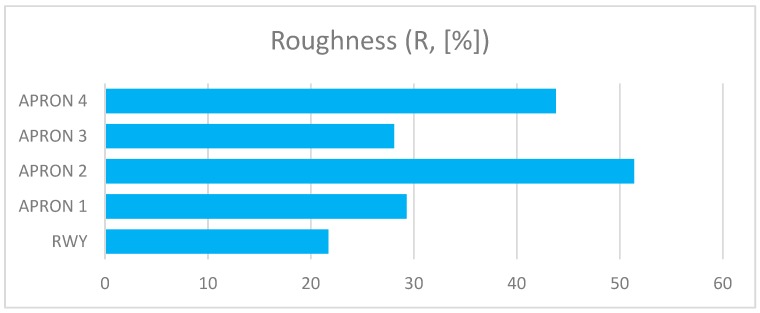
Roughness of AFE’s.

**Figure 9 ijerph-17-01663-f009:**
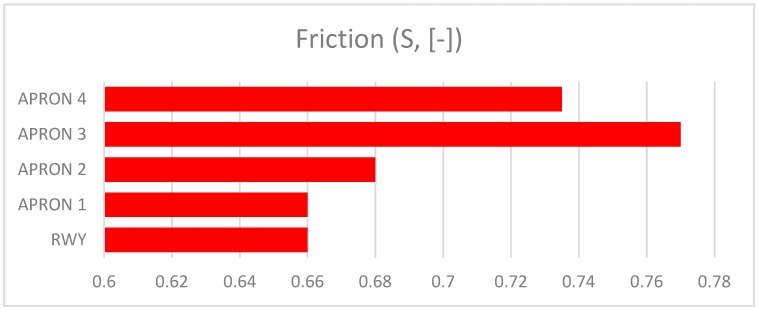
Friction of AFE’s.

**Figure 10 ijerph-17-01663-f010:**
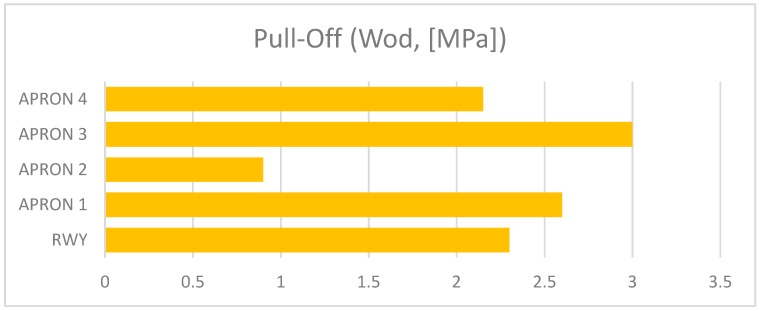
Bond strength by pull-off of AFE’s.

**Figure 11 ijerph-17-01663-f011:**
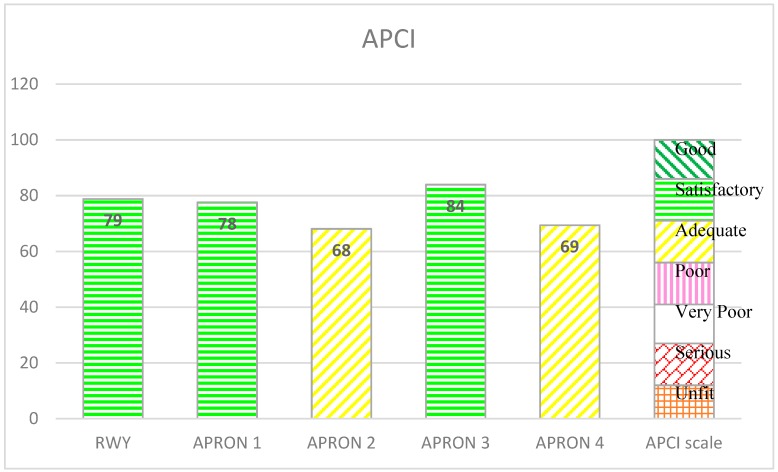
APCI evaluation of AFE’s.

**Table 1 ijerph-17-01663-t001:** Test results of in situ research.

AFE	D	PCN	R^Pd^	S	Pull-Off	APCI
	[%]	[-]	[%]	[-]	[MPa]	[-]
RWY	*10*	*39.0*	21.7	0.66	2.3	**79**
APRON 1	*10*	*26.0*	29.3	0.66	2.6	**78**
APRON 2	*19*	*34.0*	51.4	0.68	0.9	**68**
APRON 3	*8*	*49.0*	28.1	0.77	3	**84**
APRON 4	*25*	*42.5*	43.8	0.735	2.15	**69**

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
