# Peer review of "APCI Evaluation Method for Cement Concrete Airport Pavements in the Scope of Air Operation Safety and Air Transport Participants Life"

_ijerph, 2020, doi:10.3390/ijerph17051663_

Round 1

Reviewer 1 Report

1. A brief description of the findings of this study should be included in the abstract.

2. The literature review in the introduction is too rough. Besides listing relative literature, the authors should claim the links and differences between the existing papers and this work. Then the authors should give the research gaps of the research field and also present the improvements of this study. After doing these works, the contributions of this study can be verified.

3. The organization of this paper should be given at the end of the introduction section.

4. Since the method is the core of the paper, the novelty of the proposed method must be clarified in detail in Section 2.

5. The raw data including their values and units should be given. At least the authors should present part of the data for the readers to understand the inputs of the method.

6. The symbols in the model should be italic. Please revise them in the entire manuscript.

7. Are there any insights that can be revealed from the calculation results?

Above are my concerns on this paper for the author to consider. Thanks!

Author Response

 Thank you for revision. Your sugestions have been concidered and included in manuscript.

1. A brief description of the findings of this study should be included in the abstract.

Abstract was extended.

2. The literature review in the introduction is too rough. Besides listing relative literature, the authors should claim the links and differences between the existing papers and this work. Then the authors should give the research gaps of the research field and also present the improvements of this study. After doing these works, the contributions of this study can be verified.

We think that adding broader description of other mantioned methods would unnesessary expand the article.

3. The organization of this paper should be given at the end of the introduction section.

Shortyl added in the end of introduction.

4. Since the method is the core of the paper, the novelty of the proposed method must be clarified in detail in Section 2.

Novelty of proposed method where included in the end of section 2. Further information were left in section 3 as base of research result.

5. The raw data including their values and units should be given. At least the authors should present part of the data for the readers to understand the inputs of the method.

Table was added.

6. The symbols in the model should be italic. Please revise them in the entire manuscript.

Changes were made.

7. Are there any insights that can be revealed from the calculation results?

Reviewer 2 Report

In my opinion, the following remarks should be taken into account by the authors:

- (lines 6 and 7) Lack of Institutional representatives represented by the authors in accordance with the adopted formula - please add.

- (line 8) - Please add the telephone number of the author of the correspondence

- (line 9) - This statement is unnecessary - this topic is addressed by the authors at the end of the article - line 375-376.

- (line 360-374) The authors should refer in the conclusion of the article to what causes and, if so, to what extent the method proposed by them for assessing the condition of airport pavements based on the airport pavement condition indicator (APCI) differs / has an advantage over those used so far. The conclusion formulated in this way should be supported by reference to the results of the research - those presented in the article.

- (Line 375-376) the authors' contribution should be described in accordance with the adopted formula - name and initials of names and surnames. There is also no statement that "All authors reviewed the result and approved the final version of the manuscript."

Author Response

Thank you for revision.

Your sugestions were included in the manuscript, esspecially in conclusions.

Reviewer 3 Report

Overall the paper shows merit. Adding Structural testing and friction testing to PCI is good sense.  If the commonly accepted distresses are replaced with more arbitary distrsses that may be a problem.  There was little justification for using the more comprehensive testing to achive the results.

Page 2, line 51 Indianapolis is a city not a state

Page 3, figure 1 DISTRESS MISSPELLED

Page 3, line 109 add s to slabs

Page 5 Figure 3 Friction and Roughness misspelled  I don't underatand the use of the word Reparation in the context of this Figure.

Figure 5. If the graphic is a representation of time then the spacing should be 1 year not 2 years then 1 year.

Page 8 line 253 froughness should be roughness or possibly Friction

Line 220 defense standard NO-17-A501:2015 Defense standard of which nation? A photograph of this device would be useful or reference to Figure 4 PG-3.1.4

Line 220 There seems to be a problem with naming convention. Pavement roughness is not related to friction measurement or skid resistance. Pavement roughness is measured by the planograph or profileograph in US terms. Line 220 and line 253 and line 254 does not match normal convention or line 311

How does shallow flaking, deep flaking, capillaries cracks, frost cracks, marl spalls compare with ASTM D5430 distresses?

Line 284. Years of experience of whom? What entity?

Line 286 Chapter 3.1 should probably be Section 3.1

Line 294 t6hen typo

Author Response

Thank you for revision.

All sugestions were included in manuscript.

Round 2

Reviewer 1 Report

The authors addressed my concerns proposed in the first round review to a certain degree. But my last comment seems to be neglected. Moreover, I think that the revisions are a kind of simple and need further improvements. Please also avoid typos in your reply to reviewers' comments. Thanks!

Author Response

The necessary changes were made to the article, as you indicated. In fact, the last suggestion was omitted by mistake. The appropriate comment has been added in the Conclusion section. However, due to the fact that the most important part of the article is a description of the proposed APCI method and not the results of the research, the interpretation of the research results has not been studied in detail.

Round 3

Reviewer 1 Report

The second-round review is OK. Many thanks for the authors' efforts on improving their manuscript.